# Load Introduction Specimen Design for the Mechanical Characterisation of Lattice Structures under Tensile Loading

Justin Jung, Guillaume Meyer *, Matthias Greiner and Christian Mittelstedt

Konstruktiver Leichtbau und Bauweisen, Technische Universität Darmstadt, Otto-Berndt-Straße 2, 64287 Darmstadt, Germany

*   Correspondence: guillaume.meyer@klub.tu-darmstadt.de; Tel.: +49-(0)-6151-16-22025

**Abstract:** In recent years, it has been demonstrated that the lightweight potential of load-carrying structural components could be further enhanced using additive manufacturing technology. However, the additive manufacturing process offers a large parameter space that highly impacts the part quality and their inherent mechanical properties. Therefore, the most influential parameters need to be identified separately, categorised, classified and incorporated into the design process. To achieve this, the reliable testing of mechanical properties is crucial. The current developments concerning additively manufactured lattice structures lack unified standards for tensile testing and specimen design. A key factor is the high stress concentrations at the transition between the lattice structure and the solid tensile specimen's clamping region. The present work aims to design a topology-optimised transition region applicable to all cubic unit cell types that avoids high samples potentially involved in structural grading. On the basis of fulfilling the defined objective and satisfying the constraints of the stress and uniaxiality conditions, the most influential parameters are identified through a correlation analysis. The selected design solutions are further analysed and compared to generic transition design approaches. The most promising design features (compliant edges, rounded cross-section, pillar connection) are then interpreted into structural elements, leading to an innovative generic design of the load introduction region that yields promising results after a proof-of-concept study.

**Keywords:** additive manufacturing; lattice structures; load introduction design; tensile loading





## 1. Introduction

Recent trends in mechanical engineering pursue various directions, one of which is lightweight design, especially in transportation systems [1]. The reasons for this are manifold. On the one hand, it is motivated by the ecological responsibility of engineers to use fewer amounts of resources and diminish the produced emissions of machines throughout their life cycles [2,3]. On the other hand, in lightweight design, a more precise and load-specific design enables products well-suited for their applications, reducing material usage and, thus, costs [4–6]. Additive manufacturing technology is a prime example and a driving force of this trend. The main advantage of additive manufacturing resides in the unique component complexity available without increasing manufacturing costs due to reducing material usage compared to applying conventional manufacturing technologies [1].

Derived from the freedom and complexity of design offered by additive manufacturing, highly periodic structures can be advantageous and offer the highest mechanical strength and stiffness relative to their weight or volume [7,8]. This applies to lattice structures (LSs) as a sub-category of porous cellular structures. These structures present a wide range of possible applications thanks to the variety of available representative unit cells and the potential property tailoring they offer [4,8–11]. LSs enable an extreme lightweight design, which results in material, time and energy saving in fabrication and can, for example, drastically improve the strength-to-weight ratio [7].

However, the large parameter space offered by the additive manufacturing process has a high impact on the result's quality and mechanical properties, which introduces manifold optimisation opportunities but also requires thorough design [5,12–14]. These variations may lead to imperfect geometries and material connections that alter the loading conditions and influence the overall mechanical performance of LSs [15]. This issue hampers the robust integration of LSs into serial parts and explains the low number of industrial use cases [9]. Therefore, the most influential parameters need to be identified separately, categorised, classified and incorporated into the design process and modelling approaches. The engineer has the need for convenient tools to enable the comprehensive design of additive manufacturing components utilising its full potential [7]. To derive such tools and create an accurate and repeatable output, more specific and reproducible testing design methods for additive manufacturing products need to be developed [16,17]. In order to do so, the reliable testing of mechanical properties is crucial. The literature shows strong variations in the state-of-the-art practices to the extent that the measured values are difficult when compared with the predicted ones, which reduces their reliability [16–18].

In the frame of tensile testing, the connection between LSs and the test machine is essential. In the case of an inappropriate design, a local stress concentration can occur at the transition between the LSs and bulk material, which will result in an undesirable fracture at this location [19,20]. In the current literature, various designs have been developed to mitigate the stress concentration at the transition region [15]. Meyer et al. have highlighted the lack of standardised design rules for lattice tensile testing [21]. They presented a systematic design solution that proposes a transition area by means of the widely spread structural grading of strut diameters [6,17,22–29]. This stress-path-based approach satisfies both uniaxiality and stress measures but may result in large samples.

In the history of load introduction, Rankine and Maxwell were among the first to consider force flux pathways and, consequently, to set the first milestones in load-path-dependent design [30–32]. Building on this, Michell developed his prominent Michell structures [33]. Though lightweight design is not the focus of this work, the basic principles of aligning material to stress pathways are similarly relevant. More recent works demonstrate the applicability of this approach to LSs, which leads to great improvements in stiffness and strength compared to uniformly distributed unit-cell-based lattices or spatially graded lattices for the space exploitation of the design space [6,23,33]. In this context, employing topology optimisation provides superior results. By various approaches and methods, the topology optimisation method can be utilised to achieve results that lend themselves to additively manufactured lightweight structures [5] in which bio-inspired design [34,35] or lattice structures [22,25,36] can be employed. The compatibility between load-driven design and additive manufacturing is nowadays widely spread under the concept of design for additive manufacturing (or DFAM), for which topology optimisation is commonly used [7,37–41]. As highlighted by Meyer et al., the open literature generally proposes the employment of the same unit cell for both the transition and target areas in the frame of load introduction into lattice structure, while topology optimisation could provide other solutions [21].

The aim of the present contribution is to propose an alternative design solution for the lattice structures at the interface between the solid material of the tensile test specimen, which has to be clamped into the test rig, and the lattice structure. To achieve this, a novel approach combining topology optimisation, numerical design of experiment and correlation analysis is employed. Resulting from this approach, the main driving design features of load introduction in the lattice tensile specimen are extracted and interpreted into structural elements. Innovative design guidelines geared towards simple, efficient and universal sample design are derived and compared to the literature. In addition, recommendations are offered for influencing the optimisation and design variables in the frame of topology optimisation of lattice structures. These recommendations are not reported in literature to the best of the authors' knowledge and could, therefore, be useful to other researchers for further investigations. In the framework of this investigation, the

f2ccz and the bcc cubic truss unit cells are considered. First, an ample parameter study is conducted to determine the most influential and, therefore, purposeful optimisation variables and ranges by means of a numerical design of experiment (DoE) and correlation analysis. The identified parameters are then varied for further topology optimisation calculations. The attempts to design this transition region are presented and evaluated, and their deficiencies are exposed. Second, finite element analyses are conducted for the comparison of the results with generic transition design approaches. In order to facilitate the evaluation, sample criteria are formulated. Despite the configuration and load case dependency of the optimisation results, the key characteristics can be extracted and summed up into beneficial design features. These features are then combined manually to obtain a novel geometrically defined transition design proposal, the performance of which is once again assessed. Both the manufacturability and effectiveness of the new sample design are investigated in a proof-of-concept study (PoC). Finally, the draft potential towards standardisation is evaluated. In addition, the main variables driving the topology-optimised design space in the vicinity of lattice structures are identified and discussed. The potential for a unified standard for repeatable and reliable results, enabling interdependent and thorough quantification of manufacturing influences, is discussed and suggestions for future work are formulated.

## 2. Materials and Methods

In the framework of this study, the lattice structures are created using the CAD software package Siemens NX12. The models are parametrised in order to enable the simple and fast variation of the relevant geometrical parameters. The topology optimisation and subsequent finite element analyses are performed using the software package Altair Hypermesh v14.0 and the Optistruct solver. The DoE evaluations are monitored using a MATLAB script, while a PYTHON script deals with the result post-processing and plot generation.

### 2.1. Design Setup

The considered unit cells are the face-centred cubic with z-reinforcement f2ccz (Figure 1a) and the cubic body-centred bcc (Figure 1b) truss lattice structures. These unit cells are respectively representatives of the stretching- and bending-dominated structures for which different load introduction design features can be expected. The aspect ratio (AR) of the lattice structure is further employed to describe the unit cell's geometry. It consists of the relationship between the cell size a and the strut thickness t of a given cell, as shown in Equation (1).

$$\text{AR} = \frac{a}{t} \qquad (1)$$

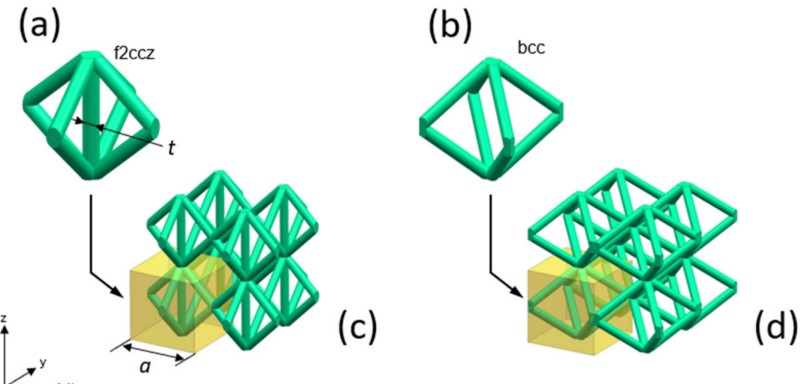

**Figure 1.** Considered unit cells and corresponding lattice structures. (**a**) f2ccz unit cell; (**b**) bcc unit cell; (**c**) f2ccz lattice structure; (**d**) bcc lattice structure.

The lattice structures are obtained from the periodic repetition of these unit cells (Figure 1c,d). The total amount of cells within a lattice structure can be determined as:

$$n_{cells,x} \times n_{cells,y} \times n_{cells,z} \qquad (2)$$

Here, n is the number of cells in the x, y or z directions, respectively.

The initial basic sample design is schematically represented for the cubic truss lattice f2ccz in Figure 2. It consists of three distinct zones: the target area, in which the desired homogeneous tensile stress state is to be ensured for a given unit cell with a constant AR; the bulk area stands for the sample bulk grip area that is used for the primary load introduction; and the transition area, which acts as a design space to guarantee a homogeneous tensile stress state at its interface with the target area. The quadratic cross-section of the bulk area in the xy-plane is directly extracted from the cubic periodicity of the selected unit cells. Therefore, the number of cells in both x and y directions is further denoted $n_{cells,xy}$. Since the detailed design of the bulk area is not part of this investigation, the sample geometry depends on the size of the considered lattice structure and the height of the investigated design space $h_D$.

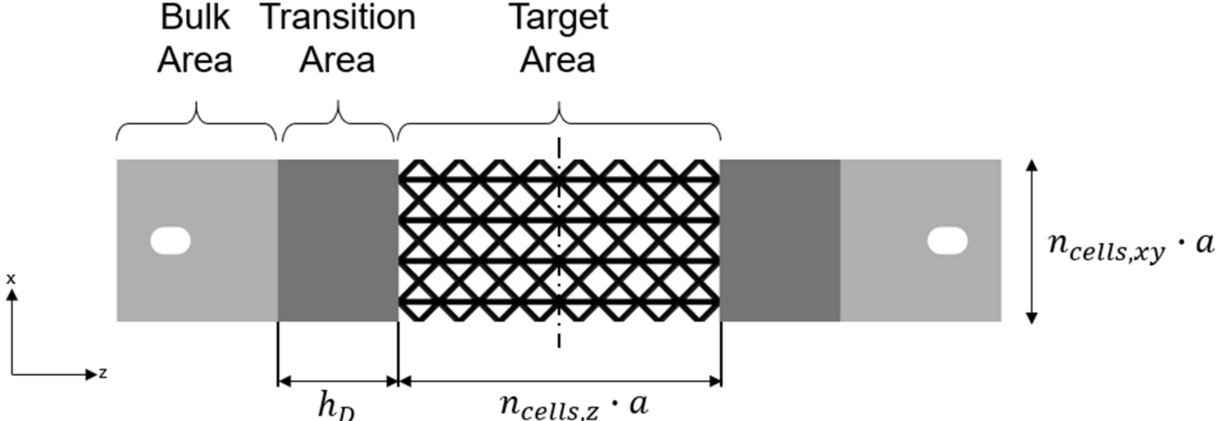

**Figure 2.** Sample design—Example of f2ccz lattice structure.

### 2.2. Numerical Setup

In the framework of the numerical investigation, both modelling and meshing approaches are employed for both the topology optimisation and finite element analyses. The optimisation-specific features are further detailed in this section.

The employed numerical model schematically depicted in Figure 3 consists of both transition and target areas. The investigated design space $h_D$ is parametrised to allow for the modelling of the lattice sample with and without a transition region. Due to symmetry considerations, the model is reduced to an eighth sample. The applied boundary conditions are defined according to the symmetry planes. The load introduction occurs at the top section of the transition area. The load introduction type is purposely not further defined at this stage because it belongs to the optimisation variables described below.

In order to obtain a precise mapping of the load distribution and corresponding stress concentration, a three-dimensional continuum model with second-order tetrahedra elements is employed [42]. A fine auto-meshing procedure is preferred over a detailed mapped meshing approach in order to reduce the computation time. After a convergence study, the mesh element size is set as 0.2 times the strut diameter. This enables the detection of not only the principal stress maxima but also local sub-maxima throughout the struts of a single unit cell.

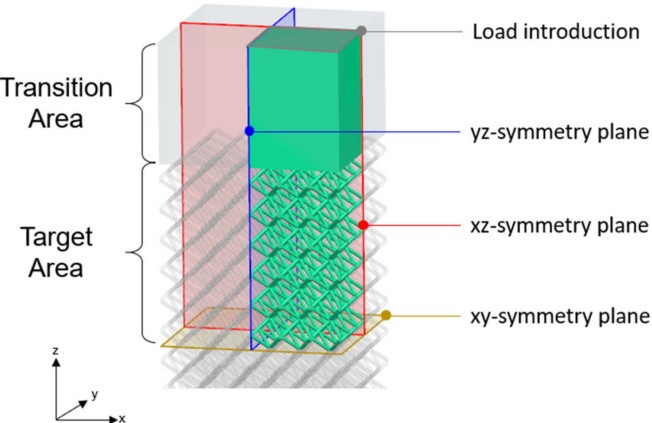

**Figure 3.** Eighth model representation.

The topology optimization setup is depicted in Figure 4. Based on former work [21] and the results of the investigation of a sample without a *transition area* (Section 3.1), two relevant areas for the application of the constraints and objectives are identified. The first critical area in a tensile sample is the top connection of the *target area*, where edge effects may take place due to the inhibition of transverse strain contraction. In the framework of this investigation, this top region covers two-thirds of the first lattice layer after the design space. Since failure shall ideally occur in the middle of the *target area*, the second optimisation area is located at the sample's centre. Here, preliminary investigations have shown that only the lowest half of the unit cells within the middle layer (nodal area excluded) are relevant to obtaining coherent results without regional ambiguity.

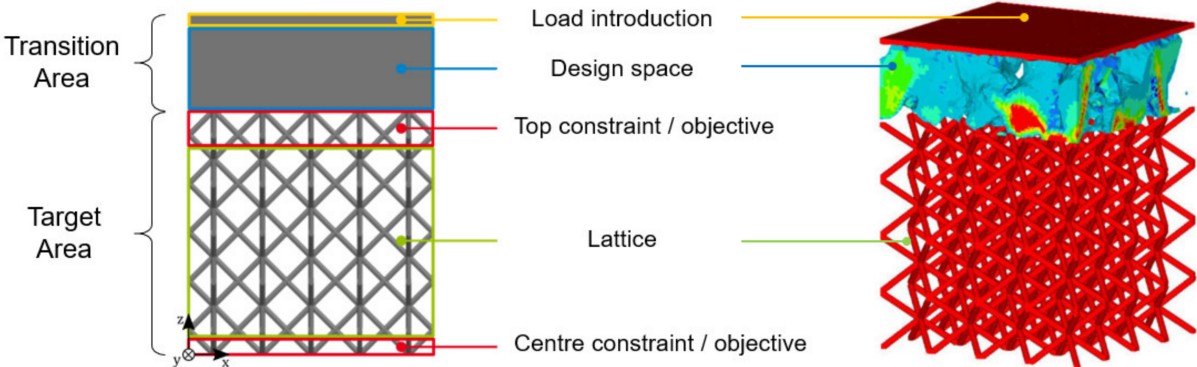

**Figure 4.** Topology optimisation setup.

Among the parameters and control cards available in Altair Hypermesh, variables prone to influencing topology optimisation results have been identified as relevant for the investigation. They are categorised as follows: geometric variables, modelling variables and optimisation variables. These variables and the ranges indicated in Equation (3) to Equation (14) are used in the frame of the numerical DoE. On the one hand, these variables are investigated to identify the design features that are independent of the specimen size or unit cell type in order to ideally reach a normalised sample specimen design. On the other hand, evaluating both the modelling and optimisation variables allows for the assessment of the influence of the numerical features on the optimisation results and, eventually, the identification of the driving parameters of the topological optimisation of lattice structures, which are not to be found in the literature to the best of the authors' knowledge.

The first geometric variable is the aspect ratio, as described by Equation (1) in Section 2.1. The investigation range takes the following features into account: the critical AR of each unit cell [8], the assumed limit between the lattice structure and porous

material [43], relevant lightweight use cases [44] and efforts involved in terms of modelling and computation. This results in the limit as follows:

$$4 \leq \text{AR} \leq 10 \tag{3}$$

Next, the geometric variables are the number of unit cells within the target area in the xy-plane and along the z-direction, which are respectively denoted as $n_{\text{cells,xy}}$ and $n_{\text{cells,z}}$. No differentiation between the x- and y-directions is made due to the cubic nature of the considered unit cells. In order to cover different sample sizes, the following ranges are considered based on the sample dimensions listed in the literature [15,21]:

$$6 \leq n_{\text{cells,xy}} \leq 16 \tag{4}$$

$$8 \leq n_{\text{cells,z}} \leq 24 \tag{5}$$

The last geometric variable is the height of the transition area $h_{\text{D}}$. The investigated design space range is directly related to the unit cell size a, although independent of it, for scaling and comparison purposes. Its upper limit has been set to reduce the sample height in comparison to a load introduction approach using graded lattice structures [21] in order to provide a significant advantage in terms of manufacturing to the proposed final design. Its lower limit is deemed to propose a realistic converging design showing the first differences when compared with a configuration without a transition area. The $h_{\text{D}}$ can be lower than a in the frame of this parameter study.

$$0.5 \cdot a \leq h_{\text{D}} \leq 6 \cdot a \tag{6}$$

The modelling variables are only restricted to the type of load applied into the transition area. The loading can be applied in three main ways, which were narrowed down to two loading cases.

The first load introduction type is an enforced displacement applied on the upper surface that is representative of real tensile tests. The applied displacement value is based on the range of displacement rates found in the literature for simulations and physical tests [8,16,17,45,46]. In the framework of this investigation, a displacement rate of 1 mm/min is chosen to justify the assumption of linear strain and, thus, not require non-linear simulations. This value is then translated into a quasi-static tensile load case, which leads to an enforced displacement of 1 mm.

The second load introduction type covers the cases of negative pressure applied to the top surface or traction force on the top nodes, which are commonly used in finite element simulations to describe the investigated load case. In the framework of this investigation, this load introduction type is simulated as negative surface pressure. For comparison purposes, the magnitude of the applied pressure is established in the frame of preliminary studies as equivalent to the applied enforced displacement of the previous load type. The values cover a range between around 0.2 N/mm$^2$ and 1.7 N/mm$^2$, depending on the stiffness of the investigated target area.

$$\text{load type} = \begin{cases} 1 \equiv U_z \\ 2 \equiv P_z \end{cases} \tag{7}$$

The following topology optimisation variables have been selected as relevant for the identification of small-scale features. For the sake of brevity, they are only explained briefly. For further details, the reader is referred to the literature [42,47,48]. In Optistruct, the parameter DISCRETE facilitates the density assignment penalty factor p in the frame of the Solid Isotropic Material with Penalization (SIMP) optimisation approach. It influences the tendency for elements in a topology optimisation to converge to a material density of 0 or 1, i.e., the tendency to assign material or not for a given element stiffness. In this study, the

suggested range of 2 to 3 has been extended. Large values can be used to identify the main load paths, while smaller ones result in more discrete structures.

$$\text{DISCRETE} = \text{p} - 1 \tag{8}$$

$$1 \leq \text{DISCRETE} \leq 4 \tag{9}$$

Similar to the penalty factor, the TOPDISC parameter can aid in the discretisation of the elements to produce further discrete results, i.e., it increases the probability of proposing filigree structures. This variable can be set to either on or off.

$$\text{TOPDISC} = \begin{cases} 0 \equiv \text{off} \\ 1 \equiv \text{on} \end{cases} \tag{10}$$

The volume fraction $V_f$ determines the remaining material infill of the design space and can, therefore, influence the topology towards thinner structure layouts. The defined goal volume fraction is set to enforce a topology with fewer material in the design space in order to focus on the main load paths.

$$0.1 \leq V_f \leq 0.5 \tag{11}$$

The minimum and maximum member size variables, respectively $m_{minmem}$ and $m_{maxmem}$, enable the elimination of either too slender or too large features in order to reach feasible design proposals. They are adapted to the strut diameter as the thinnest structural element. On the one hand, their lower limits are set to enable thin lattice-like topologies in the design space, while, on the other hand, their upper limits aim to achieve a minimal topology connection without enforcing too large overhangs.

$$0.5 \cdot t \leq m_{minmem} \leq 2 \cdot t \tag{12}$$

$$1 \cdot t \leq m_{maxmem} \leq 3 \cdot t \tag{13}$$

In the framework of this investigation, the Constraint/Objective variable is defined as a combination of the applied constraints and targeted optimisation objectives. The aim of the identified objectives is to ensure failure in the target area. In addition, in order to ensure mostly tensile stress conditions in this area, the stress constraints for a uniaxial loading have been applied. The uniaxiality condition involves restricting the principal stresses in the plane perpendicular to the loading direction $\sigma_{II}$ and $\sigma_{III}$ by negligible non-zero values $\sigma_{II,max}$ and $\sigma_{III,max}$ determined after the preliminary runs. These constraints can be applied either in the top region or in the centre region, according to the objective. The following three cases are distinguished:

1.  Minimise the maximum von Mises stress in the top lattice region $\sigma_{VM}^{Top}$ with the minimum von Mises stress constraint in the centre lattice region $\sigma_{VM}^{Centre}$ above the corresponding analytical yield stress $\sigma_{y,ana}$ of the considered unit cell [44].
2.  Maximise the minimum von Mises stress in the lattice centre region $\sigma_{VM}^{Centre}$ with stress constraints for uniaxiality in the top lattice region $\sigma_{II,max}^{Top}$ and $\sigma_{III}^{Top}$, max.
3.  Minimise the compliance of the design space $C_{hD}$ with the stress constraints for uniaxiality in the top lattice region $\sigma_{II,max}^{Top}$ and $\sigma_{III,max}^{Top}$.

$$\text{Constraint/Objective} = \begin{cases} 1 \equiv \min\left(\sigma_{VM}^{Top}\right); \sigma_{VM}^{Centre} \geq \sigma_{y,ana} \\ 2 \equiv \max\left(\sigma_{VM}^{Centre}\right); \sigma_{II;III}^{Top} \leq \sigma_{II;III,max}^{Top} \\ 3 \equiv \min(C_{hD}); \sigma_{II;III}^{Top} \leq \sigma_{II;III,max}^{Top} \end{cases} \tag{14}$$

### 2.3. DoE Setup

In the framework of this investigation, a numerical design of experiment is conducted in order to reduce the number of time-consuming runs. To do so, a Latin Hypercube Design (LHD) sampling is employed to evenly cover the multidimensional parameter space [49,50]. The input encompasses the range limits described in the previous section. Although 11 sample runs are necessary for the LHD approach, 15 samples for each lattice type are chosen as a reasonably large sample size. The sample configurations are summed up in Table 1. Additionally, this counteracts the restriction of Optistrut to require a maximum member size of at least two times the minimum member size control [42]. Therefore, the runs with too small values have turned off the maximum member size control. In such a case, the concerned variables are marked with *.

**Table 1.** Resulting LHD sample for the DoE study. Values apply for each lattice, f2ccz and bcc.

| Sample Run | AR | $n_{cells,xy}$ | $n_{cells,z}$ | $h_D$ | Load Type | DISCRETE | TOPDISC | $V_f$ | $m_{minmem}$ | $m_{maxmem}$ | Constraint/Objective |
|---|---|---|---|---|---|---|---|---|---|---|---|
| 1 | 5.12 | 10 | 12 | $1.57 \times a$ | 1 | 2 | 1 | 0.31 | $1.42 \times t$ | $1.00 \times t$ * | 3 |
| 2 | 7.84 | 10 | 8 | $3.78 \times a$ | 1 | 4 | 0 | 0.18 | $0.82 \times t$ | $0.74 \times t$ * | 1 |
| 3 | 8.98 | 6 | 4 | $2.18 \times a$ | 1 | 3 | 1 | 0.49 | $0.55 \times t$ | $0.77 \times t$ | 2 |
| 4 | 9.51 | 14 | 14 | $5.06 \times a$ | 2 | 2 | 0 | 0.25 | $0.27 \times t$ | $1.47 \times t$ | 3 |
| 5 | 6.94 | 10 | 12 | $0.97 \times a$ | 1 | 3 | 0 | 0.22 | $1.28 \times t$ | $2.13 \times t$ | 3 |
| 6 | 6.61 | 12 | 14 | $5.58 \times a$ | 1 | 4 | 0 | 0.42 | $0.67 \times t$ | $1.81 \times t$ | 1 |
| 7 | 8.34 | 6 | 18 | $3.34 \times a$ | 1 | 1 | 0 | 0.10 | $0.80 \times t$ | $1.05 \times t$ * | 1 |
| 8 | 8.49 | 14 | 20 | $3.96 \times a$ | 2 | 3 | 0 | 0.14 | $0.95 \times t$ | $1.50 \times t$ * | 2 |
| 9 | 7.56 | 16 | 8 | $5.94 \times a$ | 2 | 1 | 1 | 0.36 | $0.43 \times t$ | $1.00 \times t$ | 2 |
| 10 | 6.28 | 16 | 22 | $1.92 \times a$ | 1 | 4 | 1 | 0.32 | $0.58 \times t$ | $1.30 \times t$ | 2 |
| 11 | 9.91 | 14 | 20 | $2.64 \times a$ | 2 | 1 | 1 | 0.27 | $0.60 \times t$ | $1.11 \times t$ * | 3 |
| 12 | 4.19 | 12 | 22 | $3.14 \times a$ | 1 | 4 | 0 | 0.40 | $1.20 \times t$ | $2.16 \times t$ * | 2 |
| 13 | 4.76 | 8 | 16 | $4.52 \times a$ | 2 | 2 | 0 | 0.37 | $2.04 \times t$ | $2.36 \times t$ * | 1 |
| 14 | 5.44 | 8 | 18 | $4.88 \times a$ | 2 | 1 | 1 | 0.47 | $1.68 \times t$ | $2.40 \times t$ * | 1 |
| 15 | 5.96 | 8 | 10 | $0.75 \times a$ | 2 | 3 | 1 | 0.19 | $1.34 \times t$ | $1.72 \times t$ * | 3 |

* Maximum member size control turned off.

### 2.4. Evaluation and Visualisation of Results

Correlation matrices are employed to evaluate the results of the DoE study. The following correlation analyses are considered: the Pearson, Kendall and Spearman correlations as well as the Maximal Information Coefficient (MIC) in order to account for potential non-linear correlation. In the framework of this investigation, a threshold of significance of the calculated correlation is set for a *p*-value of 0.07. For details on these methods, the reader is referred to the literature [49,51,52]. In order to perform the correlation analyses, a set of score criteria is introduced. The structure score assesses the transferability of the topology design into structural elements (pronounced structure or design patterns). The material distribution score evaluates the allocation of material with respect to both design space and manufacturing constraints such as overhanging structures. Complementary to this, the lightweight score assesses an effective material distribution and aims at minimising the material usage, which can be relevant to the implementation of lattice structures into lightweight components. A proper load introduction is evaluated by the connectivity score, which focuses on the material connection between the transition and target areas. The stress distribution, for which the ideal case is a failure in the centre region under an almost uniaxial loading condition, is evaluated by means of finite element analyses of the resulting design topologies. The z-stress distribution score determines the load distribution in the loading direction and is compared to the xy-stress distribution score, which evaluates the uniaxiality of stresses within the whole structure. The overall score is an independent criterion that takes all the aforementioned features into account and provides an overall impression of the resulting topology on the most intuitive grading of results. These criteria are qualitatively ranked in four levels from bad to very good in order to address any score bias. All the score criteria contribute equally to the evaluation of the best runs, apart from the lightweight score and the z-stress distribution score, for which weighting factors of 0.5 and 1.5, respectively, are introduced due to their respective relevance. The final score biases

are then normalised to a range between 0 and 1 in order to ensure comparability between the employed correlation analyses.

In order to enable the quantitative stress distribution of a given sample configuration, the results are visualised for the sample's eighth, as displayed in Figure 3. As in previous work [21], the results are evaluated in three view cuts: lateral, centre and diagonal (Figure 5). The element stresses are, therefore, projected onto a two-dimensional plane. Given the three-dimensional nature of the elements and the model size, overlapping points exist. In order to keep the maxima visible, the elements are sorted for their respective stress. This ensures the visualisation of the local notch stress increase.

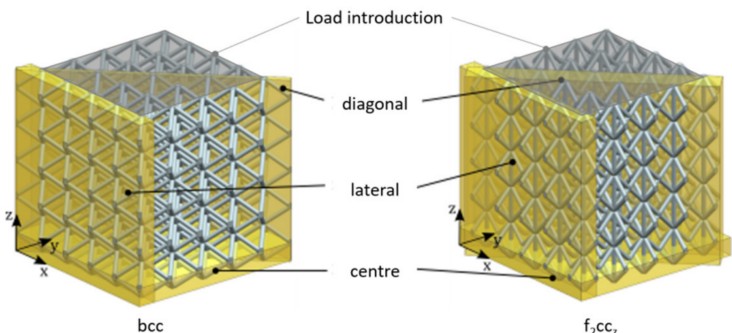

**Figure 5.** Selected view cuts.

As a qualitative aid to estimating the global stress distributions, the elements' von Mises stresses are plotted as coloured scatter plots. An underlay of the more global stress distribution is given via contour plots. Additionally, two side plots project the stress values on the respective axes, which, when combined, enable the identification of the global stress distribution and the maxima. For example, Figure 6 shows the stress distribution plot for the bcc lattice structure in the diagonal view cut. Due to the eighth model representation, the $8 \times 8 \times 8$ structure is represented here by $4 \times 4$ unit cells with the top right corner corresponding to the sample's edge.

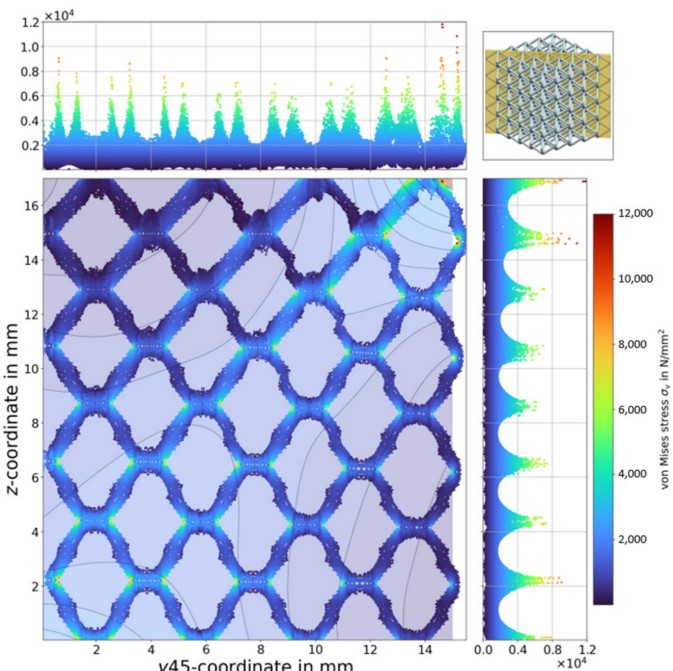

**Figure 6.** Sample without transition area—Von Mises stress of bcc $8 \times 8 \times 8$—diagonal view cut.

### 3. Results and Discussion

For the sake of brevity, only the most relevant results are discussed in this section.

#### 3.1. Samples without a Transition Region

The samples without a *transition area*, i.e., $h_D = 0$ (Figure 2), are investigated first in order to achieve a comparable results baseline. Figures 6 and 7 display the stress distribution within the bcc and f2ccz lattice samples, respectively. In both cases, the global maxima are observed in the top outer corner. The f2ccz lattice exhibits about an 15% higher maximum stress. This results from the more stretch-dominated lattice configuration, as less bending strain is enabled and the vertical strains are loaded primarily by tension. For the bcc sample (Figure 6), the local maxima are visible at the topmost elements and around the middle distance from the top of the unit cells. For the f2ccz sample (Figure 7), the topmost elements show the highest stress values as well as two additional local maxima in the vertical strut of the unit cell. In this representation, these two distinct local maxima are located on the topmost layer and below at the inner side half a unit cell from the top are visible. In both structures, the local stress concentrations can be observed in the vicinity of the nodal areas, especially in the sample's centre. However, the order of magnitude of these stress constraints is lower than the ones at the edges. This means that the structure will be more likely to fail at the corners due to edge effects, which are unwanted because they could falsify the test results. This is in line with observations made in the framework of investigations with beam elements [21]. This distribution suggests the inclusion of a region at least larger than half the top unit cell for the optimisation processes. It is visible that these stresses appear at the surface of the struts that endure the highest strain through bending. The localisation of these stress concentrations justifies the area covered for the application of the optimisation constraints and objectives at the sample's top region, as described in Section 2.2.

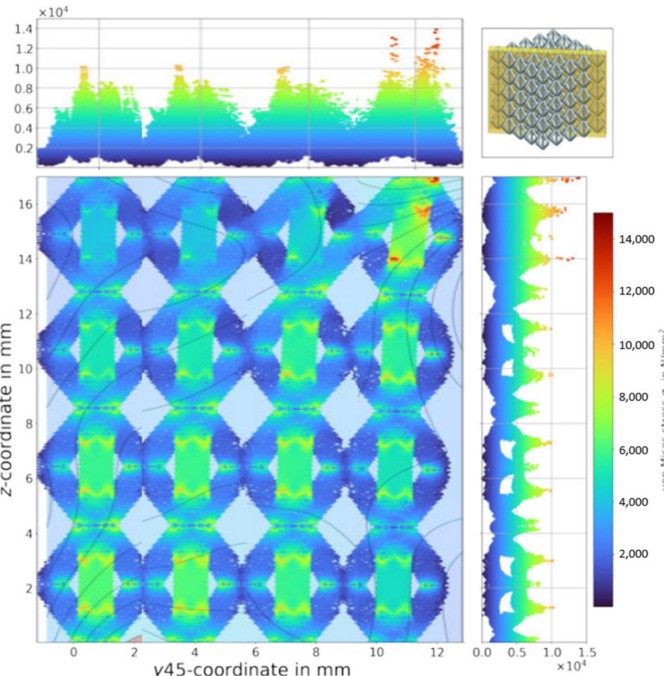

**Figure 7.** Sample without transition area—Von Mises stress of f2ccz 8 × 8 × 8—diagonal view cut.

In order to analyse the stress distribution, a displacement plot in a direction orthogonal to the loading direction is shown in Figure 8. The transverse strain distribution yields a clearly perceivable necking of the specimen, which is in line with the experimental observations by Gümrük [20] Due to the differences in stiffness between the bulk material and lattice structure, the transverse strains are locked at the top connection. The diagonal

structure of the lattices promotes the stress transfer to the centre, meaning the highest displacement in the z-direction is visible for the centre unit cells. During the elongation of the stressed specimen, the unit cells stretch and compress towards the centre, which is visible as global necking. Therefore, the highest stress is induced at the sample's edges, as they represent the farthest point from the centre for a square specimen layout. As documented by Gibson and Ashby for honeycomb and lattice structures, the initial elastic response of the stretched cells is dominated by cell-edge bending. In the stretching of the unit cells, the cell edges rotate inwards and the stiffness increases. The struts align and the deformation is dominated by stretching [43]. The described behaviour can be postulated for the f2ccz cubic unit cells, while the same behaviour can be assigned to the bcc unit cells as well, as far as single struts rather than cell edges are considered.

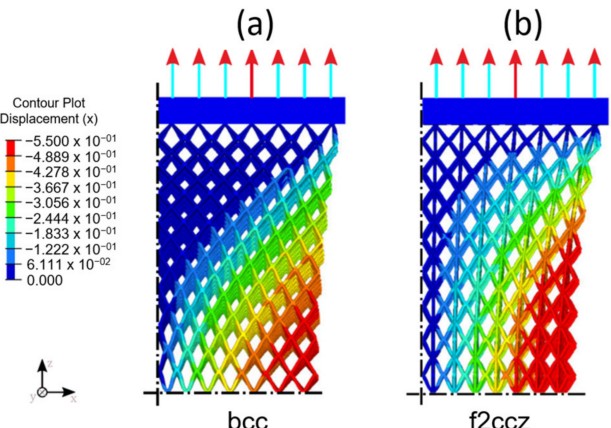

**Figure 8.** Sample without transition area—displacement in the x-direction of bcc (**a**) and f2ccz (**b**) structures—lateral view cut.

### 3.2. Optimisation Results

3.2.1. DoE Correlation Analysis

In the frame of the correlation analysis, the optimisation variables from Section 2.3 are confronted to score the criteria from Section 2.4. A summary of the main findings can be found in Table 2. Due to the statistically small number of samples investigated within the framework of this study, the outcomes of the correlation analysis are to be considered as trends rather than results leading to definitive statements. As the correlations are low, with most of the absolute values being below 0.5, particular attention was paid to the related *p*-values during the analysis of the results. The low correlation scores can be explained by the differences in the behaviours of the unit cells as well as by potential modelling precision or convergence issues, which are due to the arbitrarily broad range of optimisation variables. However, the derived results are deemed sufficient, on the one hand, to give advice on the relevant variables and corresponding ranges for further optimisation studies dealing with lattice structures and, on the other hand, to hint at common relevant structural elements.

The strongest correlations for both lattice types are observed for the number of unit cells in the plane transverse to the load direction $n_{cells,xy}$ and the type of loading. $n_{cells,xy}$ yields a direct positive correlation, which means that it is suggested to use higher unit cell numbers in the xy-plane, where possible. As discussed in the frame of a transition area based on graded lattice structures, the typical load path in truss lattice structures is three-dimensional [21]. Therefore, an ideal size for the target area can be speculated. The size can be expected to be lattice-structure-dependent and could provide different slenderness ratios for the tensile specimen. The correlation results regarding the load type highlight that a surface pressure loading should be preferred. In the preliminary studies, some displacement loaded optimisation runs failed or had no material connection from the lattice to a top connector, meaning that they were physically meaningless. Additionally, the objective/constraint combination correlates strongly for the bcc lattice but also for the

f2ccz lattice. The optimisation towards the minimisation of compliance presents higher scores. This result is in line with the convergences issues of optimisation using enforced displacement addressed in the literature [42,53–57]. In the frame of the topology optimisation of a design space in the vicinity of thin walled features such as lattice structures, and for one-dimensional loading, it is advisable to use the compliance objective with stress constraints and a loading modelled as negative pressure.

**Table 2.** Results—DoE correlation analysis—main correlations summary.

| Lattice Specificity | Variable | Correlation Outcome | Relevant Score Criteria |
|---|---|---|---|
| f2ccz | $n_{cells,xy}$ | High $n_{cells,xy}$ | General correlation |
| | load type | Negative surface pressure | General correlation |
| | $h_D$ | Low $h_D$ | Only for the z-stress distribution score |
| | Constraint/Objective | Compliance optimisation with stress constraint | General correlation |
| | $m_{minmem}$ | Low $m_{minmem}$ | Only for the connectivity score |
| | $m_{maxmem}$ | Low $m_{maxmem}$ | Only for the connectivity score |
| bcc | AR | High AR | Not high but indication through the z-stress distribution score |
| | $n_{cells,xy}$ | High $n_{cells,xy}$ | General correlation |
| | $n_{cells,z}$ | High $n_{cells,z}$ | General correlation, especially for the xy-stress distribution score |
| | load type | Negative surface pressure | Structure score and material distribution score |
| | DISCRETE | High DISCRETE | Only for the xy-stress distribution score |
| | Constraint/Objective | Compliance optimisation with stress constraint | General correlation |
| Both | TOPDISC | Turned off | No correlation observable |
| | $V_f$ | Standard value of 0.3 | No correlation observable |

Among the other investigated optimisation variables, the two important design variables that are the aspect ratio AR and the design space height $h_D$ do not reveal clear correlations. For the bcc lattice structure, the study gives an indication of a possible positive correlation with higher aspect ratios towards better stress distribution in the loading direction. As the upper limits are derived by the lattices to remain a manufacturable three-dimensional feature, medium to high aspect ratios (e.g., AR = 8) are suggested for further investigations. As no correlation for the aspect ratio in the f2ccz lattice results from the investigations, the advice for the bcc lattice can be followed too, as the results do not suggest otherwise. For the design space height and, thus, the height of the transition area, an anti-proportional correlation can be perceived for the f2ccz structure. This means that lower design space heights should be favoured to achieve an optimised stress layout. Here, it is supposed that the reduced number of design variables for the optimisation decreases the degrees of freedom for the algorithm and can, therefore, lead to more distinctive results. As a clear optimal design height cannot be obtained and no minimal transition section can be identified, it is suggested to individually adjust the height of the desired transition area of a given sample to a narrow height until a deterioration of the result is observed.

The classical optimisation parameters yield different correlation results. The minimum and maximum member size controls $m_{minmem}$ and $m_{maxmem}$ present a notable anti-proportional correlation for only the f2ccz structure. Given the feasible mesh size for the design space, the minimum member size does not necessarily affect the design to an extent that results in a different topology. If the minimum member size control is not used, a checker-board control should be applied to reduce the bad connection of the elements [42].

The maximum member size control can aid in the design but offers no distinct benefit. The suggestion is to exclude these parameters in a first run and enable them only if the specific topology material appears overly localised (enable the maximum member size control) or if no proper connection is created with the lattice (enable the minimum member size control). The DISCRETE penalty factor displays a correlation with the xy-stress distribution score for only the bcc structure. Therefore, it is suggested to use the standard values for structural problems in the case of thin-walled features too. The TOPDISC card in Optistruct shows a similar characteristic, as no considerable correlation is perceivable, and can, therefore, be disabled. The volume fraction $V_f$ shows no correlating behaviour and remains to be determined by the application, as it is mostly influenced by the desired parts weight goal. For independent optimisation, the general value of $V_f = 0.3$ can be used.

A noticeable and important combination of correlations for the following sections is observed for the structure score. This score correlates to a good extent with both the aspect ratio and constraint/objective combination variables. This means that the structural elements should be easily recognisable for the high aspect ratios and are representative of the investigated loading case, which should not encounter the aforementioned convergence problems. This highlights the trustworthiness of the results and the potential to turn the identified features into a realistic and effective design.

### 3.2.2. Relevant Topology-Optimised Design Features

The topology optimisation results of the analyses listed in Section 2.3 are visually investigated in order to identify the relevant structural elements for a sample design proposal. Although the obtained topologies are all different, similarities can be ascertained and common design features applicable to both the bcc and f2ccz lattice structures can be identified. To do so, the relevant visualisation perspectives are identified. These are, on the one hand, the bottom view of the design space for the identification of relevant structural features connecting to the lattice structure in the *target area* and, on the other hand, the isometric view for the identification of features responsible for a proper uniaxial load introduction. To establish the final design draft, the collected results are compiled into key design features. Three main key features are identified in the framework of this study. They are summed up in Figure 9, which shows samples exhibiting all the typical features for the bcc and f2ccz, respectively sample 4 (Figure 9a) and sample 10 (Figure 9b).

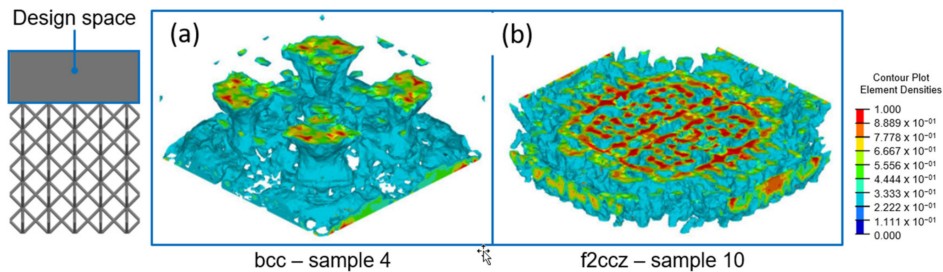

**Figure 9.** Optimised design space—bcc sample 4 (**a**) and f2ccz sample 10 (**b**).

The first identified key design feature is the presence of pillar-like structures. They can be easily identified at the connection to the unit cells for bcc samples 4, 5, 8, 11 and 15 and for f2ccz samples 4, 8, 9, 10 and 12. The strut-based unit cell types favour the pillar-like connections to introduce tensile stresses. These pillar-like structures follow the loading direction in the top connection region and ensure a direct load transfer, similar to [34]. Then, the pillar shapes depend on the investigated lattice structure because of the differences in the unit-cell-specific load paths, also similar to [21]. As can be expected, a direct pillar connection for the stretch-dominated f2ccz lattice structure is generated, since the vertical struts are aligned with the loading direction and contribute to the main load path. This design principle for tensile-load-optimised structures can also be found in other contexts for test specimens, as it enables a more uniaxial load introduction [58,59]. For the bcc

structure, the pillars are inclined, especially in the regions close to the sample's corner, for the load redistribution and load introduction into the bending-dominated lattice structure. This leads to non-circular cross-sections, and a potentially ideal design feature for load introduction in inclined struts should have an elliptical cross-section at the connection with the lattice.

The second identified key design feature is the transition from a quadratic cross-section, which is automatically implied by the unit cells, into an intermediate concentric cross-section. This cross-section can be described as circular in the specific frame of this investigation due to the applied symmetry. The best examples can be found in bcc samples 3, 4, 8, 10, 11, 12 and 15 and f2ccz samples 3, 9, 10 and 15. It should be noted that, due to the modelling approach (Section 2.2), symmetric topology results may be perceived as well as separating structures. Preliminary studies have shown few qualitative differences between the full and fourth models for significantly different computing times. This means that a potential symmetric outcome is not due to the modelling approach. The pillar-like structure is not clearly pronounced for all the bcc samples. This stems from the absence of vertical struts. In the case of the absence of pillars, the lattice top is connected via a web-like structure. Web-like connections can enable a larger vertical strain through bending. Still, these web-like structures follow a concentric pattern. In some cases, both pillar and web-like structures are combined. Although this may speak against a first intuition, since this shape does not comply with the cubic unit cell design, it can easily be understood when considering the results for samples without a transition region (Section 3.1). Circular cross-sections have no circumferential stress redirection and, therefore, no stress concentrations. A transition from a quadratic cross-section to a concentric one can reduce the edge effects and is, therefore, beneficial for the design. This finding can be compared with the full circular design of some tensile specimen attempts listed by Benedetti et al. [15]. Moreover, independent of the identified structural element, the circular-shaped pattern shows variable diameters, especially in the vicinity of the lattice structures. Thick structures are observed close to the bulk area, while smaller features are observed close to the target area, with the diameters sometimes being smaller than the one of the lattice struts themselves. This means that the stress distribution is not even, and the structurally graded features should be regarded to avoid local stress concentrations. This feature can be considered the most important design feature for a potential draft, since the concepts of both a concentric pattern and structural grading have already been proven to work in the frame of a load introduction with graded lattice structures [21].

The third and final identified key design feature is linked with the previous point, as is deals with the absence of a direct connection in the corners in the vicinity of the connection with the lattice structures. Representative examples are bcc samples 4, 10 and 13 and f2ccz samples 3, 9, 10, 13 and 14. This feature is directly linked with the avoidance of stress concentrations in the sample's corner or, in other words, edge effects. As discussed in Section 3.1, the locking of transverse strain results in a necking of the sample and a local stress concentration at the sample's corners. An elastic deformation at the connection with the lattice structure needs to be enabled to remedy this effect. Interestingly, design solutions proposing this feature yield the highest scoring results regarding both stress distribution scores independently of the design height. Therefore, it can be concluded that a large transition region is not required to achieve optimised specimen characteristics. This assessment is in line with the recommendations listed in Section 3.2.1 and provides a real advantage in comparison to a load introduction made of graded lattice structures. This third key design aspect is partially achieved by the pillar connection, as these structures can bend transversely. On the one hand, the outward inclination of the pillars observed in the bcc samples contributes to overcoming the absence of material at the sample's corner and, on the other hand, the longer pillar structures that can be locally identified offer larger deflections in a transverse direction in order to achieve more compliance. At this point, it can be deduced that all three features back each other up and it is, therefore, not meaningful to consider them separately.

In order to assess the robustness of the assessments made for the identified design features, the best ranked sample runs are analysed further. Figures 10 and 11 show exemplarily stress distributions in f2ccz sample 10 and bcc sample 10, respectively. In both cases, a shift in the stress concentration maximum from the corner to the lattice centre is observed and a mitigation of the stresses in the top corner is achieved.

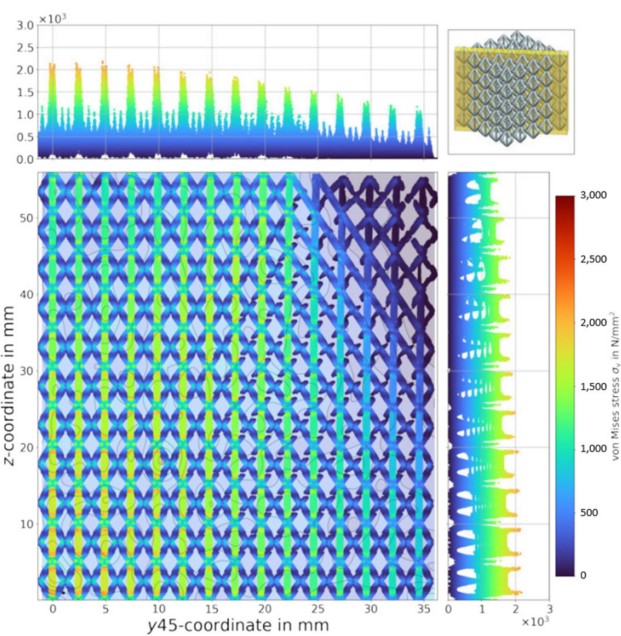

**Figure 10.** Von Mises stress of f2ccz sample 10—diagonal view cut.

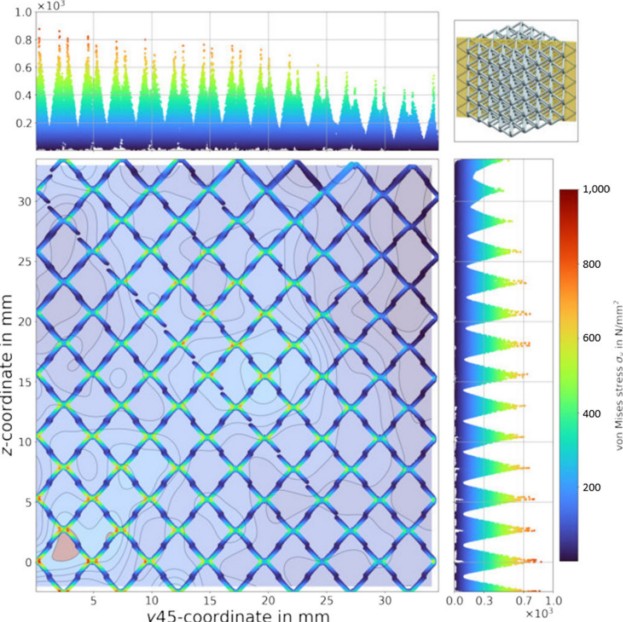

**Figure 11.** Von Mises stress of bcc sample 10—diagonal view cut.

It has to be noted here that a load introduction with different homogeneous load introduction features, such as pillars or cones, has been separately investigated and does not solve but, in the best case, only shifts the stress concentration problem. This can be linked with the abovementioned observations and the ones from Section 3.1, highlighting that a structural grading is highly relevant.

Furthermore, it can be seen that all the stress concentration issues cannot be solved with these approaches, as local peak stresses are still present at the strut junctions. Given that the local stress concentration can be reduced by local re-shaping of the lattice [24,60–62], and this issue cannot be solved by the current numerical setup, this aspect is not part of this contribution. It can be concluded that the targeted goals concerning the stress distribution identified from the analysis of a sample with a *transition area* can be successfully addressed with the proposed design measures.

*3.3. Proposed Specimen*

As mentioned in previous sections, the sample design stems from observations of repetitive features that are then interpreted into structural elements. Consequently, alternative design proposals can emerge from the observed features and be further investigated as far as they account for the aforementioned identified key features. For reasons of brevity, only the most promising design concept from Section 3.1 is shown and further investigated based on its potential universality, i.e., the transferability to other lattice structures based on cubic unit cells. The sample dimensions used in the framework of the performed numerical verification and the proof of concept not only follow the recommendations of the correlation analyses (Section 3.2.1) but also consider the ones from the literature as well. The DoE analysis hints at wide samples and, based on the observation of the design space height $h_D$, allows for a narrow transition area. The smallest investigated design space corresponding to a half of the unit cells is therefore used. This avoids unnecessarily high samples and, therefore, reduces the manufacturing time, which does not contradict the parameter trend observed in the DoE. The samples' slenderness ratio follows the recommendations from the employed norm [63] and from previous work [21]. A more detailed sample design can be investigated through either a parametric study or an analysis of both the load paths and failure modes of the lattice unit cells to be characterised, which is not the focus of this work.

3.3.1. Design

Based on the features identified in the previous section, a design proposal is made. The developed concept is applicable to both the f2ccz and bcc lattice structures and takes the restrictions of manufacturing by means of laser powder bed fusion manufacturing into account so that no supporting structures are required. No direct adjustments to the lattice need to be made. The developed concept can be used for machine connections of either quadratic or circular cross-sections. The upper half of the transition area is directly influenced by the machine connection. The design of the transition area is notched at its edges above the target area. This results in a more compliant structure that enables larger strains at the corners and, thus, reduces the stress peaks in that region. This design measure can be assimilated into the widely spread relief notch method [64]. The notch is angled so that a new stress maximum inside the transition design is avoided. In the case of a quadratic cross-section, the intermediate cross-section in the notch root presents a scalene octahedron, while a straightforward design is possible for a circular cross-section. The selected angles comply with the well-known critical inclination angle of 45° for additively manufactured parts. Two solutions for the interface between the target and transition areas are proposed. The first one involves excluding material spaces to create pillar- or alcove-like structures, whereas a good design alternative lies in the use of cones, which can be seen as graded pillars, as structural elements for proper load introduction. This solution offers the additional advantages of being compliant with DFAM approaches and providing better manufacturability, although cones have been used as space fillers rather than load introduction features [8]. The selected concept for a machine connection of circular cross-section is shown in Figure 12.

This design proposal exhibits similarities with the different tensile specimen geometries reviewed by Benedetti et al. [15] and Meyer et al. [21]. This can be explained by the presence of some of the identified key features, including compliant edges, a rounded cross-section and a pillar connection. However, none of the reported samples gather all

these driving design features at the same time. Among all the reported designs, adding a flat dog bone bulk area at the extremities of the lattice structure seems to be the least suitable design, since it can be interpreted as a sample without a transition area. This automatically implies that the sample design does not offer compliant edges, although a uniaxial loading is ensured by the design and the edge effect will take place. Hence, the sample will be more likely to fail at the interface between the bulk and lattice, especially in the case of recommended large samples. A load introduction made of lattice structures requires a load introduction design that depends on the lattice type. This requires investigating the load paths and leads to higher samples [21] than the current design proposal. A sample geometry with threaded ends corresponds to the notched design of the current proposal. In that case, the design has to be compliant with additive manufacturing for monolithic manufacturing. Although cylindrical sample shapes automatically comply with the recommendations concerning rounded cross-sections, the question of their representativity in terms of the typical load paths and corresponding comparability with analytical models of unit cells in a continuum has to be raised in the framework of the characterisation of cubic lattice structures. This question cannot be answered in the framework of the current contribution and, therefore, requires further investigation. In this regard, the design reported by Dallago et al. [65] seems the most promising alternative to be investigated and further parametrised, since the sample design considers pillar-like structures as a transition area too. In that case, it has to be ensured that the edges are compliant enough to reliably relocate failure to the sample's centre. Furthermore, a combination of a compliant transition area and graded lattice structures could represent another reliable solution in terms of the load introduction. However, this would involve more design variables and, therefore, render the design less viable and universal. Other design solutions consider a non-uniform shape on the part of the transition area, which is in line with the hints at graded structures observed in the framework of this investigation. However, the results of both the numerical verification and proof of concept show that a grading of the features in the transition area is not mandatory at this stage of the design maturity. This point represents a further optimisation possibility of the current design proposal.

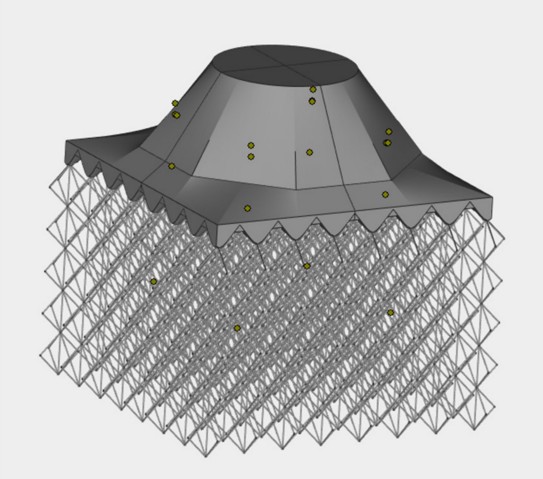

**Figure 12.** Design proposal circular machine connection with cones as the load introduction feature— example for the f2ccz lattice—isometric view.

### 3.3.2. Numerical Verification

In the framework of this investigation, numerical verification analyses of both load introduction alternatives are performed for both the f2ccz and bcc lattice structures. The results for the selected design (circular machine connection with cones) are shown in Figures 13 and 14 for the f2ccz and bcc lattices, respectively. All the observations are valid for the alternative sample designs addressed in Section 3.1. The stress distributions

clearly show an effective stress reduction on the top corners in both cases. No maxima are present in the sample's upper region, the transition area included. A stress increase towards the sample's centre is achieved for both lattices, although it is more pronounced for the f2ccz lattice, with a distinct global maximum in the centre section. The bcc maxima are distributed along the z-direction but produce a more even stress distribution in the xy-plane. The deformed shapes hint at the higher compliance of the structure in the corners when compared to Figure 8. These observations prove that the design proposal successfully fulfils its duties.

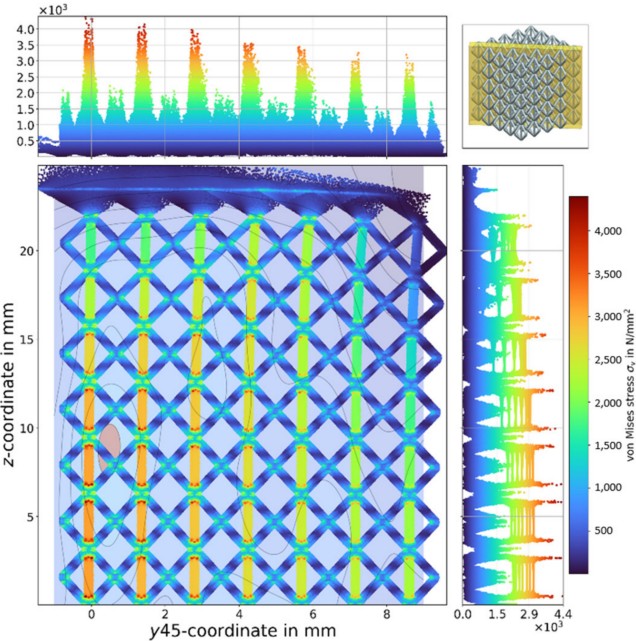

**Figure 13.** Von Mises stress of the f2ccz lattice with cones as the load introduction feature—diagonal view cut.

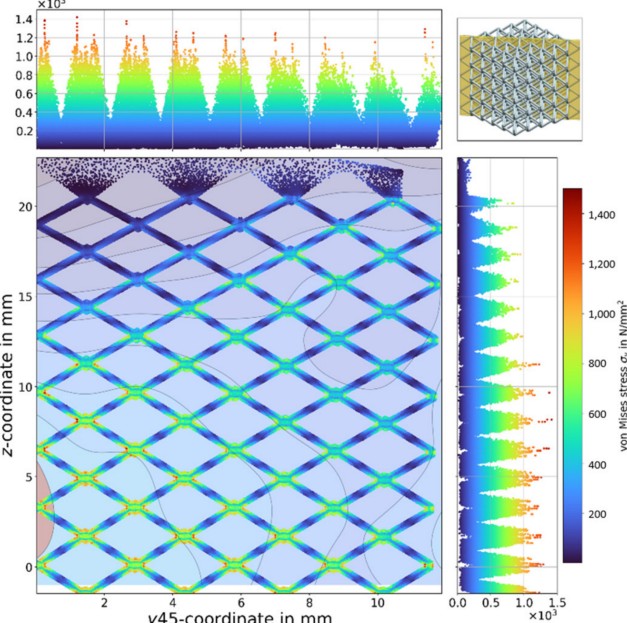

**Figure 14.** Von Mises stress of the bcc lattice with cones as the load introduction feature—diagonal view cut.

### 3.3.3. Proof of Concept Validation

The proof-of-concept validation study was deemed to be a qualitative study based on the failure scenario of the considered lattice structures. The fabrication of the specimens was carried out on a LPBF machine EOS M290 equipped with a Yb-fibre laser of an 80 μm beam diameter. For the present investigation, the commercially available powder material AlSi10Mg [66] was used. Both the layer thickness and the build plate temperature were held constant at 30 μm and 190 °C, respectively. The tensile tests were carried out on a Schenk Trebel RM600 tensile tester at the Center for Structural Materials (MPA-IfW) of the Technical University Darmstadt. The test procedure was carried out according to the standard DIN50099, which follows the concept of compression testing on cellular metals [63].

The lattice structure was assigned other parameters than the ones of the pillars and the solid load introduction. A reliable contour parameter set was selected based on the established process window for the reliable manufacturing of AlSi10Mg lattice structures published by Großmann [14]. The printed lattices were of AR = 8 and a = 3 mm, with a strut diameter t of 0.375 mm, which was achieved using a laser power of 250 W and a scanning speed of 2000 mm/s. The bulk material parts were manufactured by means of a hatch exposure strategy without a pattern that was extracted from the standard parameter sets.

The investigated structures were $10 \times 10 \times 10$ bcc and a $10 \times 10 \times 12$ f2ccz lattice structures. For each configuration, five samples were investigated. All of them showed similar behaviour and, thus, the reported samples in Figure 15 are representative for all the tensile test outcomes.

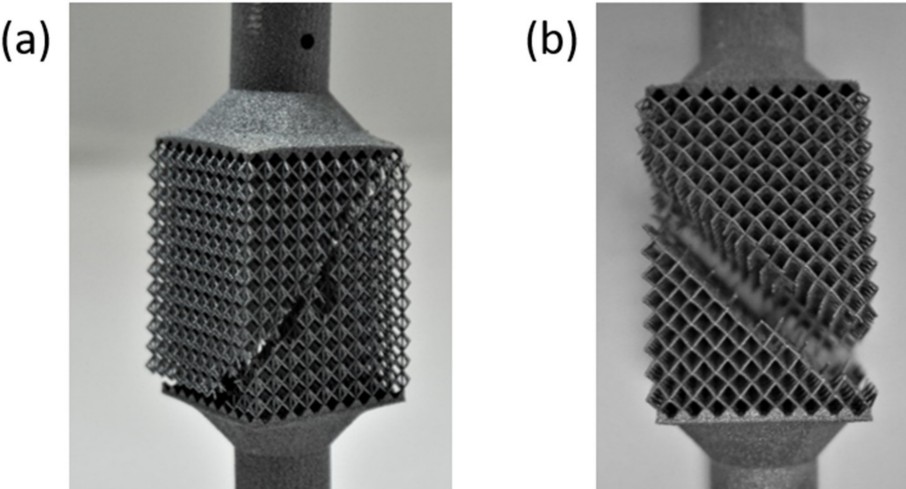

**Figure 15.** Failed specimen: f2ccz (**a**), bcc (**b**).

No major imperfections resulting from the design were visible. Both the lattice structures showed the typical fracture pattern of stretching- and bending-dominated lattice structures, respectively. Both displayed shear band failures, which developed after a load redistribution into the lattice struts in the vicinity of the initially failed one. It was observed that the failure path of the f2ccz specimens failed at the interface to the transition area, while this was not the case for the bcc specimens. Similar to [21], it can be supposed that the typical load and, therefore, failure paths depend on the unit cell considered. To prevent a failure of the f2ccz lattice structure in the vicinity of the transition area, the slenderness ratio of the tensile specimen should be increased. However, finding an ideal sample size is not the focus of this work. These promising results need to be deepened in further parametric studies.

3.3.4. Potential towards Normalised Design Guidelines

Despite the number of samples run and the potential inaccuracy of the mesh or convergence problems, the derived topologies of the DoE study present structural aspects that can be rebuilt in CAD models. The topologies themselves do not provide a direct design draft for a standardisation. However, the derived design aspects that present a beneficial characteristic for tensile specimens give a suggestion for a standardised design. With deeper physical tests pending, the proposed geometry is merely a first attempt at a proof-of-concept design, which can be improved on. The evaluated topologies can support generically driven design towards simpler specimen designs. These designs may be parameterised and adjusted to create different scales. For example, the cross-section of the *transition area* may be further thinned and shaped to be more circular, the corner notches may be adjusted for a desired strain rate and the overall height can be reduced, saving in this way material and build time. It is highly expected that the design can be transferred to other cubic unit cells due to them not necessarily being strut-based. Further transferability towards non-cubic lattice unit cells is feasible as far as the samples without transition regions display similar issues in terms of the stress concentrations. In the case of deviating challenges, first insights into relevant topology optimisation variable have been provided. Finally, other design features of the topology optimisation results can be used to create similar designs. A foundation for a normative design is given.

## 4. Conclusions

In the present work, the state-of-the-art regarding the mechanical testing of lattice structures was evaluated and deficiencies exposed. As a crucial aspect, the transition between the bulk and target areas has been identified. The local stress concentration needs to be reduced, while the maximum yield stress should occur in the centre of the specimen, ideally induced by an uniaxially distributed loading. A design of experiments study with various topology optimisations has been conducted to identify influential variables with respective ranges. Based on the results, the influential optimisation parameters for topology optimisations regarding f2ccz and bcc lattices have been assessed and show the potential to be transferred to similar cubic lattice structures. The evaluation of the correlation between the topology optimisation parameters and the resulting properties allowed us to identify important parameters that can be used for further simulations of the cubic lattice structures. The main outcomes of the correlation analysis results can be summed up as follows:

- Small lattice structures, i.e., with few unit cells, should be avoided. This enables a three-dimensional load distribution along the main load paths that are typical of a given unit cell.
- A small transition area should be preferred, since it reduces the sample height as far as a load introduction without stress concentration is guaranteed. It is advisable to individually adjust the height of the desired transition area of a given sample to a narrow height until a deterioration of the result is observed.
- In the framework of the topology optimisation analysis of lattice structures under tensile loading, it is advisable to use the compliance objective with stress constraints and a loading modelled as negative pressure.
- The optimisation variables identified as relevant for the identification of small-scale features can be set as standard first. It is advisable to enable them only if the specific topology material appears overly localised or if no proper connection is created to the lattice.

The topologies were evaluated and the geometrical key features for a desired stress condition in tensile specimens were exposed. The conducted topology optimisations did not provide an optimal solution to the problem themselves. However, the method of topology optimisation in general has proven to be able to aid in the development of a structure by highlighting important design features. From these main features, universal design guidelines can be extracted. An optimal design for tensile specimens for the characterisation of lattice structure should include the following features:

- Compliant edges to avoid edge effects;
- Concentric cross-section at the interface with the bulk area;
- Pillar-like and/or web-like structures.

These three design measures should be implemented together in a sample design, since all three result in a shift in the stress concentration maximum from the corner to the lattice centre and a mitigation of the stresses in the top corner.

Based on the insights derived from the topology optimisations, an LPBF compatible design proposal for an optimised transition structure was introduced. Promising proof-of-concept validation tests introduced the proposed design as a robust draft for standardised samples for the investigation of lattice structures under tensile loading. The proposed design is, thus, suggested for further detailed research.

**Author Contributions:** Conceptualisation, G.M.; methodology, G.M. and J.J.; software, J.J.; validation, G.M. and M.G.; formal analysis, J.J.; investigation, G.M., J.J. and M.G.; resources, C.M.; data curation, J.J.; writing—original draft preparation, G.M., J.J. and M.G.; writing—review and editing, G.M. and C.M.; visualisation, G.M., J.J. and M.G.; supervision, G.M. and C.M.; project administration, C.M. All authors have read and agreed to the published version of the manuscript.

**Funding:** This research received no external funding.

**Data Availability Statement:** Data are available on request.

**Acknowledgments:** The authors wish to thank Marius Hofmann (Dipl.-Ing) and the Center for Structural Materials (MPA-IfW) of the Technical University of Darmstadt for providing their test facility, the test execution and the transmission of the generated data.

**Conflicts of Interest:** The authors declare no conflict of interest.

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
