# Peer review of "Load Introduction Specimen Design for the Mechanical Characterisation of Lattice Structures under Tensile Loading"

_jmmp, doi:10.3390/jmmp7010037_

Round 1
Reviewer 1 Report
In this work, the authors proposed alternative design solutions to the transition area made of lattice structures. They incorporated a topology-optimized transition structure between the solid material of the tensile test specimen, as well. The manuscript is well prepared; however, the following comments can help improve the manuscript:
· The novelty of the work is not clearly highlighted.
· The authors need to discuss how they figured out the hd, n(cells, x,y).a and n(cells, z).a for the Sample design when they designed a set-up in Figure 2.
· The authors are recommended to review https://doi.org/10.1007/s11665-021-05894-y and https://doi.org/10.1108/RPJ-03-2022-0087 for the second and third paragraphs of the introduction section.
· In most of the Figures, the authors are recommended to use (a) and (b) rather than (left) and (right).
· The conclusion section needs to provide more detailed information, quantitative rather than qualitative.
Author Response
The authors are thankful for the careful consideration of the submitted article and for the constructive comments.
- The novelty of the work is not clearly highlighted.
Based on this and other reviewer’s comment, abstract, introduction and conclusion were updated so that the novelty of the work is highlighted.
- The authors need to discuss how they figured out the hd, n(cells, x,y).a and n(cells, z).a for the Sample design when they designed a set-up in Figure 2.
As mentioned in section 3.3.4, concrete figures cannot be identified from the DoE results. As mentioned from line 611 of the initially submitted manuscript on, sample dimensions follow the recommendations of the correlation analyses as well as the ones from literature. For the special case of the design space height hd, the smallest investigated height of a = 0.5 x a is used - based on the correlation analysis. This avoids unnecessarily high samples and therefore reduces manufacturing time. A detailed sample design needs to be investigated in a parametric study (as suggested in section 3.3.4), which is not the focus of this initial work.
These discussion points are now implemented at the beginning of section 3 for clarity. Besides, the investigation range of n_(cells,xy) and n_(cells,z) is now justified in section 2.2
- The authors are recommended to review https://doi.org/10.1007/s11665-021-05894-y and https://doi.org/10.1108/RPJ-03-2022-0087 for the second and third paragraphs of the introduction section.
In light of the reviewer’s comments, the proposed papers have been reviewed and the first contribution is now cited.
- In most of the Figures, the authors are recommended to use (a) and (b) rather than (left) and (right).
In light of the reviewer’s comments, changes have been applied (not marked).
- The conclusion section needs to provide more detailed information, quantitative rather than qualitative.
The conclusion section has been updated. Main findings and takeaway messages are now included.
Reviewer 2 Report
In the present work the state-of-the-art regarding mechanical testing of lattice structures was evaluated and deficiencies have been exposed. This new insight is important. I recommend publishing it as its present form.
Author Response
Many thanks for the positive feedback. The manuscript has been carefully revised according to the comments raised by the other two reviewers.
Reviewer 3 Report
this article presents a work that employs topology optimization, identifies influential parameters in additive manufacturing as an innovative design strategy to improve the mechanical properties of printed samples. The work presents the context of application and introduces well the topic. Methodology and experiments are conducted and explained appropiately. article well written, good conclusions. broad,relevant and recent research literature.
Author Response

(The authors gave the same response as above.)

Reviewer 4 Report
The paper entitled “Load introduction specimen design for the mechanical characterization of lattice structures under tensile loading” numerically investigates the design approach for transitions zones in lattice structures. The topic is interesting, but there are some points which would require improvements:
1. Line 70 and 80: too many reference are cited together: the authors should describe each individual contribution in a specific way;
2. Line 150: “Parametrized” should be replaced with “Parametrised”;
3. The section should not indicated in each figure caption: for example, at line 130 “Figure 1. Design setup – Considered unit cells and corresponding lattice structures.” should be replaced with “Figure 1. Considered unit cells and corresponding lattice structures.”;
4. Fig 4 and Fig. 10 are not clear;
5. Despite the need for brevity illustrated by the authors, the parameters investigated must be clearly described in the present work, while only the literature (for further details) should be referred to the reader. For example, what are the DISCRETE, penalty factor p and TOPDISC parameters?
6. The innovativeness of the work is not clear, moreover the analysis presented too qualitative (as the results reported in tab.2). The proposed design (line 550) has been obtained through a specifical methodology or only by authors observations? Observing Fig. 5 it seems that the stress is more concentrated at the center of the lattice structure, so the comparison should be made also in terms of mechanical resistance of the specimen with the same load.
7. Line 633: the authors state that: “The results for the bcc samples are considered as sufficient for the experimental validation.”. According to what the authors say this? The results obtained from the experimental tests and their contribution to the analysis are unclear.
Author Response
The authors are thankful for the careful consideration of the submitted article and for the constructive comments.
- Line 70 and 80: too many references are cited together: the authors should describe each individual contribution in a specific way
The references are now separated from each other so that each one of them can be justified independently.
- Line 150: “Parametrized” should be replaced with “Parametrised”
In light of the reviewer’s comments, changes have been applied (not marked).
- The section should not indicate in each figure caption: for example, at line 130 “Figure 1. Design setup – Considered unit cells and corresponding lattice structures.” should be replaced with “Figure 1. Considered unit cells and corresponding lattice structures.”
In light of the reviewer’s comments, changes have been applied (not marked).
- Fig 4 and Fig. 10 are not clear
Both figures have been reworked. Besides, the caption of figure 4 has been updated (not marked).
- Despite the need for brevity illustrated by the authors, the parameters investigated must be clearly described in the present work, while only the literature (for further details) should be referred to the reader. For example, what are the DISCRETE, penalty factor p and TOPDISC parameters?
The role of the aforementioned parameters is now described.
- The innovativeness of the work is not clear, moreover the analysis presented too qualitative (as the results reported in tab.2). The proposed design (line 550) has been obtained through a specific methodology or only by authors observations? Observing Fig. 5 it seems that the stress is more concentrated at the centre of the lattice structure, so the comparison should be made also in terms of mechanical resistance of the specimen with the same load.
Based on this and other reviewer’s comment, abstract, introduction and conclusion were updated so that the novelty of the work is highlighted.
As mentioned in section 3.2.2 and especially in line 441, the sample design stems from observations of repetitive features. These features are then interpreted into structural elements leading to the sample design proposal. The core of the design methodology is the DoE described in section 2.3. Since a DoE is employed, the results of this analysis are deemed as representative. The observed features can therefore be considered as specific. An interpretation based on observations is not fully specific, so other design proposal can emerge from the observed features. However, there is not much design freedom left when considering the restrictions given by additive manufacturing, the machine connection and the lattice unit cells (which have been considered – see section 3.1). Other design proposals have been made internally, based on observations too. The criterion of selection is the universality of the design; i.e. transferability to other lattice structures. A detailed sample design needs to be investigated in a parametric study (as suggested in section 3.3.4), which is not the focus of this initial work. These discussion points are now implemented into section 3 for clarity. Besides, the end of section 3.3.1 now includes a comparison between the proposed design and samples reported in literature.
Figure 8 clearly shows stress peaks at the sample’s edges although high stresses can be observed in the sample’s centre too. This has been observed in the framework of investigations with beam elements too (Ref 21 of the revised manuscript). Further stress concentration areas for samples without transition area are now discussed in section 3.1.
- Line 633: the authors state that: “The results for the bcc samples are considered as sufficient for the experimental validation.”. According to what the authors say this? The results obtained from the experimental tests and their contribution to the analysis are unclear.
We have removed this statement since it is not entirely clear and does not contribute to the understanding of the paper.
Reviewer 5 Report
Manuscript discussed about the load introduction specimen design for the mechanical characterization of lattice structures under tensile loading.
1. Please evaluate the abstract and emphasise the novelties, significant discoveries, and conclusions.
2. The primary theoretical and practical contributions of this study to this field of study should be described by the authors.
3. The analysis of the results is restricted to comparing the experimental findings. The authors are advised to compare and contrast the findings of their study with the body of existing literature.
Author Response
The authors are thankful for the careful consideration of the submitted article and for the constructive comments.
- Please evaluate the abstract and emphasise the novelties, significant discoveries, and conclusions.
The abstract has been reworked according to the reviewer’s comment while accounting for the word limit.
- The primary theoretical and practical contributions of this study to this field of study should be described by the authors.
Based on this and other reviewer’s comment, abstract, introduction and conclusion were updated so that the novelty of the work is highlighted.
- The analysis of the results is restricted to comparing the experimental findings. The authors are advised to compare and contrast the findings of their study with the body of existing literature.
Since it is extremely seldom to find reported failure of tensile lattice specimens, only a comparison with existing sample designs is relevant in the authors’ opinion. Differences with samples design as found in literature are now addressed in section 3.3.1. In this regard, a new reference has been added ([65]).
Round 2
Reviewer 4 Report
I am satisfied that the authors responded adequately to the reviewers' comments. In my opinion the revised paper in its current form is acceptable for publication in the Journal.